# Robust Image Self-Recovery against Tampering using Watermark Generation with Pixel Shuffling

## Abstract

The rapid growth of Artificial Intelligence-Generated Content (AIGC) raises concerns about the authenticity of digital media. In this context, image self-recovery, reconstructing original content from its manipulated version, offers a practical solution for understanding the attacker's intent and restoring trustworthy data. However, existing methods often fail to accurately recover tampered regions, falling short of the primary goal of self-recovery. To address this challenge, we propose ReImage, a neural watermarking-based self-recovery framework that embeds a shuffled version of the target image into itself as a watermark. We design a generator that produces watermarks optimized for neural watermarking and introduce an image enhancement module to refine the recovered image. We further analyze and resolve key limitations of shuffled watermarking, enabling its effective use in self-recovery. We demonstrate that ReImage achieves state-of-the-art performance across diverse tampering scenarios, consistently producing high-quality recovered images.

## 1 Introduction

Recent advances in Artificial Intelligence-Generated Content (AIGC) (Betker et al., 2023; Blattmann et al., 2023; Esser et al., 2023; Liu et al., 2024; Luo et al., 2023) have led to models capable of producing content nearly indistinguishable from real data. While these advancements offer significant benefits in fields such as media, entertainment, and digital communication, they also pose serious risks. The ease of generating hyper-realistic modified content has raised concerns about the unintentional spread of misinformation and potential ethical and legal challenges. These risks are particularly concerning in cases where manipulated media—such as forged news or altered legal evidence—can distort public perception, influence opinion, or undermine societal trust.

To mitigate this risk, prior efforts have focused on detecting or localizing tampered regions. Among these, watermarking-based image tamper localization methods (Sander et al., 2025; Zhang et al., 2024b) have shown promise by embedding watermark into images that can later indicate whether tampering has occurred. While such methods are helpful for interpreting the attacker's intent or enabling partial reuse of trustworthy regions, they do not provide access to the original image. As a result, it remains challenging to assess the significance of altered regions or the extent of semantic shifts from the original content.

Image self-recovery (Cao et al., 2024b; Ying et al., 2021; 2023) offers a promising alternative by aiming to reconstruct the original content from its tampered version. These methods embed information about the target image into itself as a watermark, allowing the original content to be recovered through watermark extraction. This enables not only a better understanding of the attacker's intent but also the complete reuse of the restored content. However, existing image self-recovery techniques suffer from notable limitations. They often fail to accurately reconstruct the tampered regions—the core objective of self-recovery—resulting in blurry restorations, which lack fine-grained structural and semantic details. This limits their utility in high-fidelity applications.

In this work, we propose ReImage, a self-recovery framework based on neural watermarking, where a shuffled version of the original image is embedded into itself as a watermark. To ensure high visual fidelity and accurate reconstruction, we design a generator that produces watermarks opti-

mized for neural watermarking. This design induces misalignment between tampered regions and their corresponding watermark locations, allowing corrupted areas to be replaced with intact counterparts and further refined via an image enhancement module. We also analyze the limitations of shuffling, where increased high-frequency components in the watermark degrade both visual quality and extraction accuracy. By addressing this, we enable its practical use in neural watermarking. ReImageachieves state-of-the-art performance across diverse tampering scenarios, and consistently produces high-quality recovered images. In summary, our overall contributions are:

- We analyze key limitations of shuffled watermark embedding in neural watermarking and enable its practical use previously infeasible in prior works.
- We introduce a novel framework that leverages a watermark generator and an image enhancement module to achieve high-fidelity image self-recovery.
- The proposed model achieves state-of-the-art performance across diverse tampering scenarios in image self-recovery.

## 2 RELATED WORKS

**Neural Watermarking**     Neural watermarking aims to imperceptibly embed data within cover content for secure, authenticated, or traceable media transmission. Traditional watermarking techniques (Chan & Cheng, 2004; Chantrapornchai et al., 2014; Wong et al., 2009) embed watermark in either the spatial or frequency domains, but often face limitations in capacity and robustness. With the advancements in deep learning, recent approaches such as Tree-Rings (Wen et al., 2023), RoSteALS (Bui et al., 2023), and others (Tang et al., 2019; Zhao et al., 2024) have achieved improved embedding capacity and robustness. Invertible Neural Networks (INNs) (Dinh et al., 2015) provide bijective mappings that enable precise data embedding and extraction, making them well-suited for neural watermarking. HiNet (Jing et al., 2021) pioneered INN-based neural watermarking with a high-capacity approach, while ISN (Lu et al., 2021a) proposed a method to embed images within images, using a key for extraction. We adopt neural watermarking for self-recovery. Unlike prior methods that embed external or fixed watermarks, our approach embeds a shuffled version of the target image itself, enabling redundant encoding of image content for self-recovery.

**Image Self-Recovery**     Image self-recovery aims to recover the original image from a tampered version, even under various manipulation attacks such as inpainting and splicing. Traditional image self-recovery methods (Cao et al., 2024a; He et al., 2006; Molina et al., 2019) are based on Least Significant Bit (LSB) embedding. However, they are highly susceptible to common degradations such as JPEG compression, and often fail to recover the original content when the tampering rate is high. Imuge (Ying et al., 2021) introduces a self-embedding strategy, where the original image is embedded into itself to reconstruct an original image. Imuge+ (Ying et al., 2023) improves reconstruction quality by adopting an invertible neural network architecture. However, both approaches limit their robustness against unseen attack types and low recovery quality in tampered region. The W-RAE (Cao et al., 2024b) method addresses this issue by embedding a $2 \times 2$ shuffled version of the original image, enabling the replacement of tampered regions with their intact counterparts. However, $2 \times 2$ shuffling fails to recover when the attacked region also covers its corresponding shuffled area. In our work, we address this issue by applying shuffling with a finer grid, made possible through our novel watermark generation and image enhancement techniques for robust recovery.

## 3 METHODS

Image self-recovery aims to restore a tampered or corrupted image to its original, authentic from tampering attack (Betker et al., 2023; Blattmann et al., 2023). This restoration process is performed without relying on any external reference or manual intervention. By facilitating the reconstruction of the original image, this process helps in understanding the attacker's intent, evaluating the extent of tampering, and ultimately enhancing trust in the authenticity of the content. In this work, we focus on watermarking-based approaches, where the information necessary for self-recovery is embedded into the original image as an invisible watermark. Specifically, let $I_{\text{org}}$ denote the original image. We embed information, derived from the original image $I_{\text{org}}$, into itself for self-recovery and produce the container image $I_{\text{con}}$. This container image may subsequently undergo tampering or corruption, resulting in an attacked image $I_{\text{att}}$. The goal of image self-recovery is to learn a function $f$ that

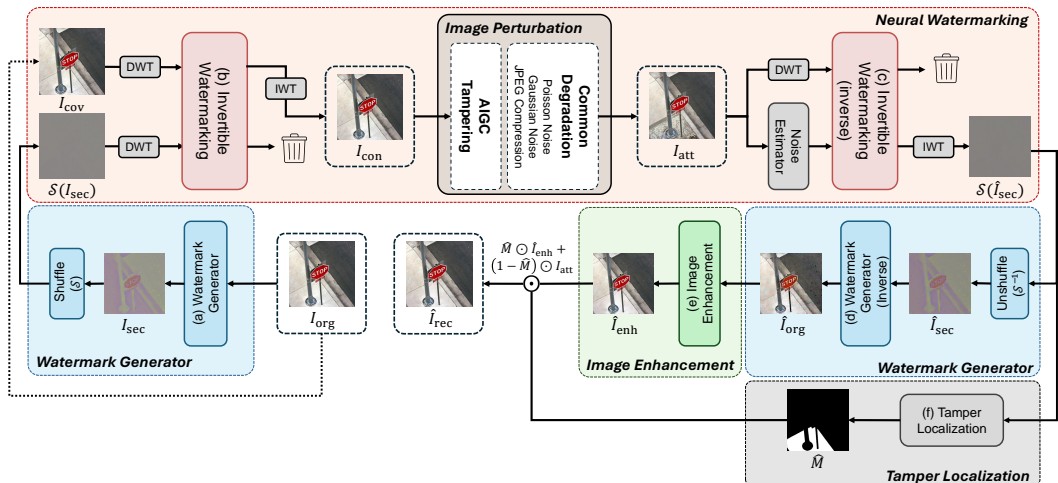

Figure 1: **Overall architecture of the proposed methods**. In our framework, given a target image $I_{\text{org}}$, module (a) generates a secret image $I_{\text{sec}}$ optimized for watermarking. The image is shuffled to obtain $\mathcal{S}(I_{\text{sec}})$, which is then embedded into the cover image $I_{\text{cov}}$—identical to $I_{\text{org}}$—via module (b), yielding the container image $I_{\text{con}}$. An attacker may generate a tampered image using an AI tool, resulting in the attacked image $I_{\text{att}}$. During image self-recovery, Noise Estimator predicts the discarded noise component $\hat{I}_{\text{noise}}$, which is used in module (c) to recover the shuffled secret image $\mathcal{S}(\hat{I}_{\text{sec}})$ from $I_{\text{att}}$. This image is then unshuffled to obtain $\hat{I}_{\text{sec}}$ and passed through module (d) to reconstruct the target image $\hat{I}_{\text{org}}$. Module (e) refines $\hat{I}_{\text{org}}$ using contextual information, producing an enhanced recovered image $\hat{I}_{\text{enh}}$. Finally, module (f) locates tampered regions and restores them using corresponding untampered areas from $I_{\text{att}}$, yielding $\hat{I}_{\text{rec}}$.

reconstructs the original image from the attacked image, *i.e.*, $I_{\text{org}} = f(I_{\text{att}})$, where $f$ denotes the self-recovery function.

### 3.1 PRELIMINARY: INVERTIBLE NEURAL NETWORK

Invertible Neural Networks (INNs) (Dinh et al., 2016) are architectures that consist of invertible blocks with the unique property of reversibility, enabling the exact recovery of inputs from their outputs. Specifically, given inputs $X_1$ and $X_2$, an invertible block maps them to outputs $Y_1$ and $Y_2$, and exactly recovers the inputs via the inverse mapping. At layer $l$, the forward process takes input $X_1^l$ and $X_2^l$, and produces $X_1^{l+1}$ and $X_2^{l+1}$ as follows:

$$X_1^{l+1} = X_1^l + \phi(X_2^l), \tag{1}$$

$$X_2^{l+1} = X_2^l \odot \exp(\sigma(\rho(X_1^{l+1}))) + \eta(X_1^{l+1}), \tag{2}$$

where $\phi$, $\rho$, and $\eta$ are neural networks, $\sigma$ is a sigmoid activation function, and $\odot$ denotes element-wise multiplication. The inversion operation does not require explicit inversion of the internal networks $\phi$, $\rho$, and $\eta$. Instead, the original inputs can be precisely reconstructed from their outputs using the following invertible formulations:

$$X_2^l = (X_2^{l+1} - \eta(X_1^{l+1})) \odot \exp(-\sigma(\rho(X_1^{l+1}))), \tag{3}$$

$$X_1^l = X_1^{l+1} - \phi(X_2^l). \tag{4}$$

An INN with $L$ blocks takes an input pair $X_1$ and $X_2$, designated as $X_1^0$ and $X_2^0$, and produces an output pair $X_1^L$ and $X_2^L$. The original $X_1$ and $X_2$ can then be recovered from these outputs through the inverse process.

### 3.2 NEURAL WATERMARKING WITH INN

Image watermarking is the process of invisibly embedding a *secret* image, referred to as the watermark, into a *cover* image, resulting in a *container* image. The embedded watermark in the container

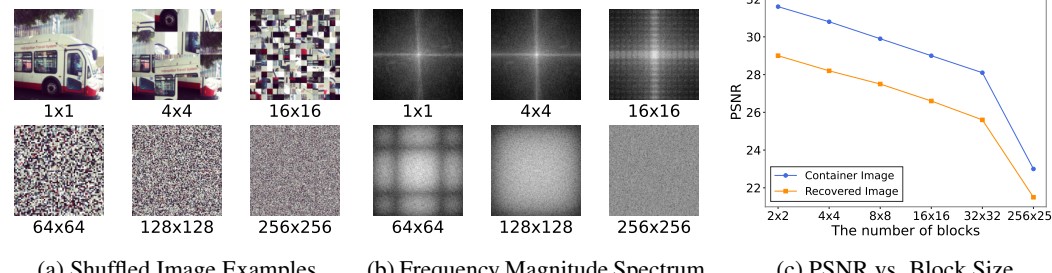

(a) Shuffled Image Examples     (b) Frequency Magnitude Spectrum     (c) PSNR vs. Block Size

Figure 2: **Effects of Pixel Shuffling.** (a) Visualization of pixel-shuffled images with different grid configurations, defining how the image is divided into equally sized patches: finer grids introduce higher variance between neighboring pixels, indicating increased high-frequency content. (b) Magnitude spectra of the images in (a), obtained via FFT: a brighter center indicates dominant low-frequency content, suggesting a smoother image. As the patch size decreases, spectral energy spreads outward, indicating an increase in high-frequency components. (c) PSNR of container and recovered images under varying the grid configurations: both degrade significantly as the grid becomes finer, due to the increased high-frequency component.

must be extractable later, enabling applications such as copyright protection and tampering detection. Due to its reversibility, the INN is well-suited for this task, allowing both watermark embedding and extraction within a unified framework. Specifically, given a cover image $I_{\text{cov}}$ and a secret image $I_{\text{sec}}$, the INN takes both as input and generates a container image $I_{\text{con}}$ as follows:

$$I_{\text{con}}, I_{\text{noise}} = \text{IW}(I_{\text{cov}}, I_{\text{sec}}), \tag{5}$$

where IW denotes the INN-based watermarking module composed of $L$ invertible blocks. After this process, the extra output $I_{\text{noise}}$ is discarded. The resulting container image $I_{\text{con}}$ stays visually similar to the cover image $I_{\text{cov}}$ while invisibly containing the information of the secret image $I_{\text{sec}}$. In inverse process, IW extract $I_{\text{cov}}$ and $I_{\text{sec}}$ from the tampered container image $I_{\text{att}}$ as follows:

$$\hat{I}_{\text{cov}}, \hat{I}_{\text{sec}} = \text{IW}^{-1}(I_{\text{att}}, \hat{I}_{\text{noise}}), \tag{6}$$

where $\text{IW}^{-1}$ is the inverse process of IW. Here, $\hat{I}_{\text{noise}}$ is an estimate of the discarded noise $I_{\text{noise}}$, which is not available for watermark extraction in practice. We adopt the network architecture from Zhang et al. (2024a) for this noise estimator to predict $\hat{I}_{\text{noise}}$ from the attacked image $I_{\text{att}}$. Similarly, the internal sub-networks $\phi$, $\rho$, and $\eta$ at layers of IW are implemented following Huang et al. (2017). Specifically, each subnetwork consists of five convolutional layers with a kernel size of $3 \times 3$, followed by Leaky ReLU.

For training, we enforce visual consistency between $I_{\text{con}}$ and $I_{\text{cov}}$ by applying an $L_2$ loss combined with the VGG loss (Zhang et al., 2018) $\mathcal{L}_{\text{LPIPS}}$, which measures perceptual similarity in VGG feature space. The overall watermarking loss is defined as $\mathcal{L}_{\text{W}} = \|I_{\text{con}} - I_{\text{cov}}\|_2^2 + \lambda \mathcal{L}_{\text{LPIPS}}(I_{\text{con}}, I_{\text{cov}})$. Additionally, to recover the secret image $I_{\text{sec}}$, we define the extraction loss $\mathcal{L}_{\text{E}} = \|I_{\text{sec}} - \hat{I}_{\text{sec}}\|_2^2$.

Since our goal is self-recovery, we may set the original image $I_{\text{org}}$ as both the cover $I_{\text{cov}}$ and the secret $I_{\text{sec}}$, resulting in a container image $I_{\text{con}}$. The output $I_{\text{con}}$ closely resembles $I_{\text{org}}$ while embedding hidden redundant information necessary for image self-recovery. As prior work (Lu et al., 2021b) has demonstrated the effectiveness of the above image watermarking, this self-embedding approach may seem sufficient to recover the original image when the container is corrupted. However, recent work (Zhang et al., 2024a) has shown that INNs exhibit fragility and locality properties. In particular, corrupted regions in the attacked image $I_{\text{att}}$ tend to align with missing content in the extracted secret image $\hat{I}_{\text{sec}}$, making accurate recovery challenging in those corrupted regions.

### 3.3 SHUFFLED WATERMARK GENERATOR

#### 3.3.1 PIXEL SHUFFLING AND UNSHUFFLING

In self-recovery, restoring corrupted regions is essential for reconstructing the original content, as the uncorrupted areas remain intact. However, due to the fragility and locality of the IW, tampering

affects corresponding regions in both the container image $I_{\text{con}}$ and the embedded secret $I_{\text{sec}}$, often causing recovery to fail. To address this, we intentionally disrupt this spatial alignment. Specifically, instead of directly using the secret image, we apply a predefined pixel shuffling algorithm $\mathcal{S}$ to obtain a shuffled version $\mathcal{S}(I_{\text{sec}})$, which is used as the watermark. As a result, the corrupted regions in $I_{\text{sec}}$ are no longer spatially aligned with those in the container. We apply the unshuffling algorithm $\mathcal{S}^{-1}$ to rearrange $\mathcal{S}(\hat{I}_{\text{sec}})$, which is the recovered watermark, into the original spatial configuration of $I_{\text{sec}}$. In this process, the missing information in $\hat{I}_{\text{sec}}$ is likely to differ from that in the container image $I_{\text{con}}$, meaning that information lost in $I_{\text{con}}$ may still be present in $\hat{I}_{\text{sec}}$.

**Shuffling vs Shifting** To induce spatial misalignment between corrupted regions in the container image and those in the secret image, shifting may be an alternative option. However, shifting becomes less effective when the inverse shifting is applied to the reconstructed secret image, where tampering typically takes the form of bulk attacks. In the case of shifting, the corrupted region remains spatially clustered even after the inverse transformation. This makes recovery difficult, as the lack of surrounding uncorrupted pixels limits the available cues for reconstruction—contradicting the objective of restoring tampered regions accurately. In contrast, with shuffling, mapping each pixel back to its original location disperses the majority of the corruption throughout the entire image, thereby providing sparse cues for reconstruction across the whole image.

**Shuffling and Watermarking Quality** Shuffling also exhibits some challenges. As shown in Figure 2, applying fine-grained pixel shuffling intensifies high-frequency components in image, which are clearly visible in Figure 2a. To visualize this effect, we present the magnitude spectra obtained from the FFT across different block sizes (Figure 2b), thereby confirming the increase in high-frequency components induced by finer shuffling. This phenomenon leads to a noticeable degraded visual quality of both the container and recovered images, as shown in Figure 2c. This is because a higher proportion of high-frequency components reduces compressibility, making it more difficult to embed the watermark imperceptibly. Similar trends have also been observed in Cao et al. (2024b).

### 3.3.2 WATERMARK GENERATION

To address the visual quality degradation caused by high-frequency dominance, we propose a Watermark Generator (WG) composed of stacked INN blocks, which transforms the original image into a secret image optimized for neural watermarking, as follows:

$$I_{\text{sec}} = \text{WG}_1(I_{\text{org}}, I_{\text{org}}) \tag{7}$$

where $\text{WG}_1$ refers to the first output component of WG. The secret image $I_{\text{sec}}$ is subsequently shuffled using the predefined shuffling process $\mathcal{S}$. Given a fixed shuffle, WG is trained to generate a image suitable for watermarking after the shuffling. Hereafter, we refer to $I_{\text{sec}}$ as the output of WG. Since WG is implemented using an INN block, we can approximately reconstruct the original image $I_{\text{org}}$ through inverse transformations as follows:

$$\hat{I}_{\text{org}} = \text{WG}_1^{-1}(\hat{I}_{\text{sec}}, \hat{I}_{\text{sec}}) \tag{8}$$

where $\text{WG}_1^{-1}$ refers to the first output component of $\text{WG}^{-1}$ and $\hat{I}_{\text{sec}}$ is obtained by unshuffling extracted secret image $\mathcal{S}(\hat{I}_{\text{sec}})$. We adopt transformer encoder (Vaswani et al., 2017) for the internal subnetworks $\phi$, $\rho$, and $\eta$ within each INN block. Each transformer encoder block processes image patches of size $P$ using GELU activations, resulting in a feature map of spatial dimensions $\mathbb{R}^{\frac{H}{P} \times \frac{W}{P}}$. To restore the original resolution $\mathbb{R}^{H \times W}$, two transposed convolutional layers are subsequently applied. This architecture captures long-range dependencies across spatial positions.

We regularize WG to generate a smooth and high-frequency suppressed watermark after applying the shuffle operation by reducing spatial variation between neighboring pixels, which is expressed by the following Total Variation loss (Rudin et al., 1992):

$$\mathcal{L}_{\text{TV}} = \sum_{i=1}^{H-1} \sum_{j=1}^{W-1} \left[ \left( \mathcal{S}(I_{\text{sec}}^{(i,j)}) - \mathcal{S}(I_{\text{sec}}^{(i,j+1)}) \right)^2 + \left( \mathcal{S}(I_{\text{sec}}^{(i,j)}) - \mathcal{S}(I_{\text{sec}}^{i+1,j}) \right)^2 \right] \tag{9}$$

where $i$ and $j$ denote pixel indices and $H$, $W$ are the image height and width. This loss penalizes abrupt changes between adjacent pixels, resulting in a high-frequency suppressed $\mathcal{S}(I_{\text{sec}})$. Additionally, we introduce a reconstruction loss $\mathcal{L}_{\text{WG}} = \|I_{\text{org}} - \hat{I}_{\text{org}}\|_2^2$ to ensure that the predicted original image $\hat{I}_{\text{org}}$ remains close to the original image $I_{\text{org}}$.

### 3.4 IMAGE ENHANCEMENT

Since we apply the shuffling algorithm, the recovered image $\hat{I}_{\text{org}}$ exhibits globally distributed degradation. To address this, we propose an Image Enhancement (IE) module that leverages contextual information from $\hat{I}_{\text{org}}$ to improve its perceptual quality. Specifically, we employ an IE module, which refines $\hat{I}_{\text{org}}$ by utilizing the sparsely reconstructed regions, producing the enhanced recovered image $\hat{I}_{\text{enh}}$. To implement this module, we adopt the architecture from Liu et al. (2025), which was originally developed for the super-resolution. As a result, the recovered output exhibits significantly improved perceptual and structural quality. To ensure the enhanced recovered image $\hat{I}_{\text{enh}}$ closely resembles the original image $I_{\text{org}}$, we apply an $L_2$ loss combined with the VGG loss $\mathcal{L}_{\text{LPIPS}}$:

$$\mathcal{L}_{\text{IE}} = ||\hat{I}_{\text{enh}} - I_{\text{org}}||_2^2 + \lambda \mathcal{L}_{\text{LPIPS}}(\hat{I}_{\text{enh}}, I_{\text{org}}). \tag{10}$$

### 3.5 TAMPER LOCALIZATION

During inference, we generate the final output $\hat{I}_{\text{rec}}$ by selectively combining the enhanced image $\hat{I}_{\text{enh}}$ with the attacked image $I_{\text{att}}$, in order to preserve untampered content and enhance overall visual quality. Accordingly, we employ a Tamper Localization (TL) module to identify manipulated regions. We leverage the inherent fragility and locality of the IW, where tampering causes corruptions in the embedded secret image, regardless of the attack method. This enables model-agnostic tamper localization based on the output of the IW. To leverage these properties, we adopt a U-Net-based TL, inspired by Zhang et al. (2024a), to generate pixel-wise tampering masks $\hat{M}$. The final output is computed based on the predicted mask as $I_{\text{rec}} = \hat{M} \odot \hat{I}_{\text{enh}} + (1 - \hat{M}) \odot I_{\text{att}}$. To train the tamper localization module, we apply a pixel-wise binary cross-entropy loss between the ground-truth tampering mask and the predicted mask. This loss is denoted as $\mathcal{L}_{\text{TL}}$.

### 3.6 TRAINING

**Common Degradation** To enhance robustness, we introduce random common degradations—such as Gaussian Noise, JPEG Compression, Gaussian Filter and Median Filter—applied to the container image during training. These transformations simulate real-world degradations that may unintentionally affect the container image. Although they preserve the semantic content, they alter image quality, thereby improving the model's robustness to such distortions.

**Random Masking Strategy** We simulate attacks by applying random masking strategies during training. Specifically, we replace a randomly sampled region, covering 10% to 50% of the entire image, with a random image selected from the dataset. We use two different shapes for the random region, as described in Sander et al. (2025). Additional details are provided in the Appendix.

**Final Loss Function** The total loss is defined as the weighted sum of all loss components:

$$\mathcal{L}_{\text{total}} = \lambda_{\text{W}} \mathcal{L}_{\text{W}} + \lambda_{\text{E}} \mathcal{L}_{\text{E}} + \lambda_{\text{TV}} \mathcal{L}_{\text{TV}} + \lambda_{\text{WG}} \mathcal{L}_{\text{WG}} + \lambda_{\text{IE}} \mathcal{L}_{\text{IE}} + \lambda_{\text{TL}} \mathcal{L}_{\text{TL}}. \tag{11}$$

## 4 EXPERIMENTS

### 4.1 EXPERIMENTAL SETTINGS

**Dataset** We use MS-COCO2017 (Lin et al., 2014) for both training and evaluation. For training, we utilize the images from the regular train split of Lin et al. (2014), which contains 118K images, without any annotations. For evaluation, we adopt the valAGE-Set split with 1,000 images, as proposed in Zhang et al. (2024a). This subset is drawn from the regular validation set. Segmentation annotations from this subset are used as target regions for simulating tampering with AIGC models. (Stable Diffusion Inpaint (SD-Inpaint) (Rombach et al., 2022), Stable Diffusion XL (SDXL) (Podell et al., 2023) and Splicing). To assess the robustness of the models, we apply a random combination of Gaussian Noise, JPEG Compression and Poisson Noise during the evaluation.

**Evaluation Metrics** To assess the quality of both container and recovered images, we employ Peak Signal-to-Noise Ratio (PSNR), Structural Similarity Index Measure (SSIM), and Learned Perceptual Image Patch Similarity (LPIPS). These metrics are commonly used to compare the original image with the container and recovered images (Sander et al., 2025; Ying et al., 2023; Zhang et al., 2024b).

Table 1: **Impact of Core Components in ReImage.** "PS", "IE" and "WG" indicate Pixel Shuffling, Image Enhancement module and Watermark Generator, respectively. Using all components yields the best performance in terms of the visual quality of both container and recovered images. M-PSNR denotes the PSNR of the recovered image computed only over the tampered regions.

| Methods | PS | IE | WG | Container Image $I_{con}$ | | | Recovered Image $\hat{I}_{rec}$ | | | |
| | | | | PSNR↑ | SSIM↑ | LPIPS↓ | PSNR↑ | SSIM↑ | LPIPS↓ | M-PSNR ↑ |
|---|---|---|---|---|---|---|---|---|---|---|
| (a) | ✗ | ✗ | ✗ | 35.93 | 0.93 | 0.10 | 25.82 | 0.85 | 0.18 | 16.44 |
| (b) | ✓ | ✗ | ✗ | 25.78 | 0.64 | 0.40 | 22.95 | 0.58 | 0.44 | 18.32 |
| (c) | ✓ | ✓ | ✗ | 26.19 | 0.65 | 0.35 | 24.25 | 0.65 | 0.37 | 18.91 |
| (d) | ✓ | ✓ | ✓ | 36.10 | 0.93 | 0.10 | 30.57 | 0.87 | 0.17 | 23.93 |

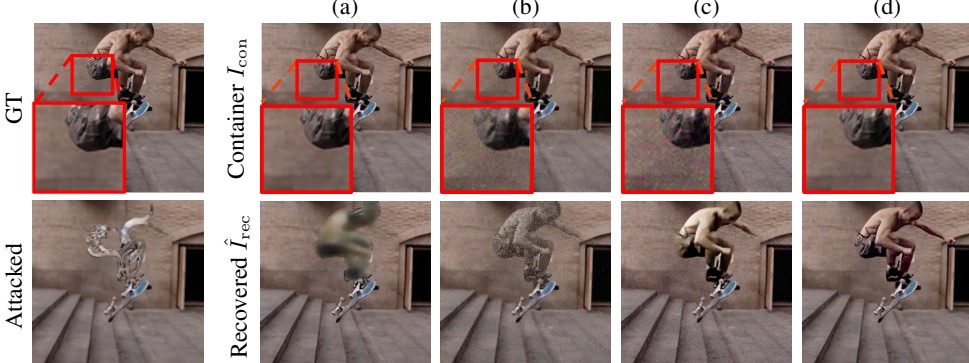

Figure 3: **Qualitative Comparison across Component Configurations.** We qualitatively analyze the visual impact of each core component. The settings (a) to (d) correspond to those in Table 1. Tampering is simulated using SD Inpaint, and the red box indicates the region that is magnified for closer inspection. As components are progressively added, the recovery of tampered regions improves. While settings (a), (b), and (c), which exclude the WG, exhibit visible artifacts in the container image, setting (d) produces a cleaner result that closely resembles the ground truth.

**Implementation Details** We employ 12 invertible blocks for IW and 3 for WG, using a patch size of $P = 4$. All models are trained for 200K iterations using the Adam optimizer (Kingma & Ba, 2014) with a learning rate of $2 \times 10^{-4}$, $\beta_1 = 0.9$, and $\beta_2 = 0.5$. The input image resolution is fixed to $256 \times 256$. The loss weights are configured as follows: $\lambda = 10$, $\lambda_W = 150$, $\lambda_E = 10$, $\lambda_{TV} = 10$, $\lambda_{WG} = 10$, $\lambda_{IE} = 20$, and $\lambda_{TL} = 1$. During training, tampered regions are excluded from the computation of $\mathcal{L}_E$ so that only the uncorrupted areas contribute to the loss.

## 4.2 RESULTS

**Results in Ablation Study** Table 1 shows the ablation results analyzing the contributions of each core component in our method: Pixel Shuffling (PS), Watermark Generator (WG) and Image Enhancement (IE) module. All models are trained independently using only the target components along with the TL module. For all experiments, we simulate tampering using Stable Diffusion Inpaint (SD Inpaint) (Rombach et al., 2022). In addition to the standard metrics, we also report M-PSNR, which measures PSNR specifically within the tampered regions. This metric is particularly focused on the reconstruction of the manipulated area, a key objective of the self-recovery task.

When embedding the input image into itself using the IW module (a) from Table 1, the container maintains good quality, while the recovered image exhibits significantly lower quality. Furthermore, when measuring the recovered quality within the tampered region alone, the scores decline even further. This degradation is due to the pixel alignment between the container and the secret image, as discussed in Section 3.3.1. With this alignment, the specific region we aim to recover—namely, the tampered region in the container—becomes significantly disrupted shown by low M-PSNR. This is also qualitatively observed in Figure 3, where the recovered image within the tampered region appears blurry with limited information.

Table 2: **Comparison of Different Self-Recovery Methods on valAGE-Set.** Tampering is simulated using three methods: SD Inpaint (Rombach et al., 2022), SDXL (Podell et al., 2023), and splicing following Hou et al. (2024). To evaluate the visual quality of the recovered and container images, we use PSNR, SSIM, and LPIPS. † We retrained W-RAE with common degradations to ensure fair comparisons, as the original model, which was trained without them, exhibited poorer recovery performance.

| Models | Container Image $I_{con}$ | | | Recovered Image $\hat{I}_{rec}$ | | | | | | | | |
|---|---|---|---|---|---|---|---|---|---|---|---|---|
| | | | | SD Inpaint | | | SDXL | | | Splicing | | |
| | PSNR↑ | SSIM↑ | LPIPS↓ | PSNR↑ | SSIM↑ | LPIPS↓ | PSNR↑ | SSIM↑ | LPIPS↓ | PSNR↑ | SSIM↑ | LPIPS↓ |
| Imuge | 31.23 | 0.89 | 0.02 | 22.32 | 0.62 | 0.27 | 22.76 | 0.64 | 0.26 | 22.15 | 0.63 | 0.26 |
| Imuge+ | 32.38 | 0.77 | 0.17 | 23.59 | 0.64 | 0.23 | 22.55 | 0.64 | 0.23 | 24.26 | 0.63 | 0.25 |
| W-RAE | 38.04 | 0.96 | 0.01 | 21.58 | 0.56 | 0.36 | 18.53 | 0.48 | 0.42 | 21.61 | 0.57 | 0.36 |
| W-RAE† | 29.27 | 0.76 | 0.18 | 23.61 | 0.63 | 0.34 | 23.63 | 0.63 | 0.34 | 23.08 | 0.63 | 0.35 |
| ReImage | 36.10 | 0.93 | 0.10 | 30.57 | 0.87 | 0.17 | 30.59 | 0.87 | 0.17 | 29.75 | 0.87 | 0.17 |

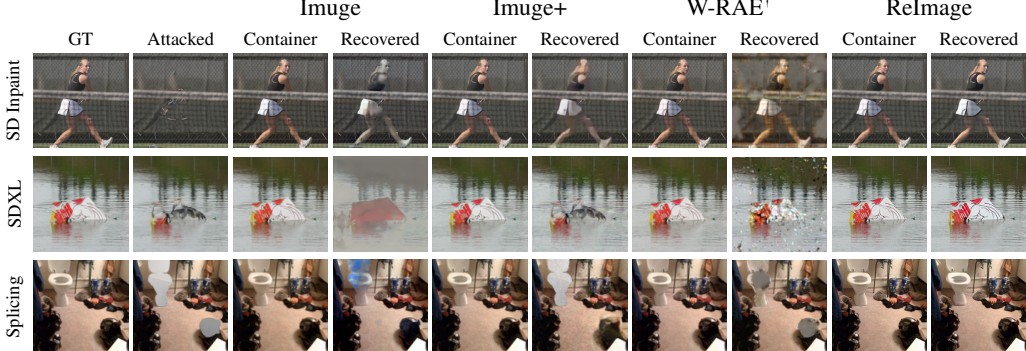

Figure 4: **Qualitative Results of Different Self-Recovery Methods.** Visual tampering is simulated as outlined in Table 2. We compare our method with Imuge, Imuge+, and W-RAE†, showing that our approach yields clearer and more complete restorations.

When PS is applied (b), both the container and recovered images show degraded visual quality, compared to (a). This degradation results from the dominant high-frequency components within the watermark introduced by shuffling, as discussed in Section 3.3.1. However, the higher M-PSNR compared to (a) suggests that the tampered regions are better reconstructed thanks to the misalignment between the container and the secret images. The qualitative example in Figure 3 demonstrates increased noise in both the container and recovered images, while the original structure of the corrupted region is more clearly reconstructed, though still sparsely.

Applying IE (c) improves the visual quality of the recovered image by filling the dispersed broken regions by leveraging the sparse contextual information. The container image also benefits, as IE helps to restore missing details, enabling IW to embed less information while still enabling effective recovery. The improved M-PSNR compared to (b) demonstrates the effectiveness of this module. Qualitatively, it reduces noise in the tampered regions.

Finally, WG (d) significantly enhances the quality of both the container and recovered images, achieving relative gains of 38% and 26% in PSNR, respectively, compared to (c), by optimizing the watermark for image self-recovery. The notable 26% improvement in M-PSNR highlights the substantial contribution of the quality enhancement in the tampered region to the overall improvement in the recovered image. Qualitatively, the magnified regions of the container image, (b) and (c) exhibit noticeable noise, whereas (d) closely resemble the GT. The recovered image also demonstrates effective restoration of fine details, indicating the fidelity of the proposed approach. For completeness, results for all component combinations are provided in the Appendix.

**Comparisons to SOTA Methods for Image Self-Recovery** We compare our method against recent state-of-the-art self-recovery approaches, including Imuge (Ying et al., 2021), Imuge+ (Ying et al., 2023), and W-RAE (Cao et al., 2024b). Tampering is simulated using Stable Diffusion Inpaint (SD Inpaint) (Rombach et al., 2022), Stable Diffusion XL (SDXL) (Podell et al., 2023), and splicing adopted from Zhang et al. (2024a;b), guided by mask annotations from the valAGE-Set. Since W-

Table 3: **Comparison of Self-Recovery Performance under Various Common Degradations.** We evaluate robustness of models by measuring the quality of recovered images under six common degradation types: Gaussian Noise (G.N.), JPEG Compression (JPEG), Gaussian Filter (G.F.), Median Filter (M.F.), Poisson Noise (P.N.), Hue Adjustment, Brightness Adjustment and Contrast Adjustment, along with "Clean" where no degradation is applied. Note that † indicates models retrained with these degradations.

| Metrics | Methods | Clean | G.N. | JPEG | G.F. | M.F. | P.N. | Hue | Brightness | Contrast |
|---------|---------|-------|------|------|------|------|------|------|------------|----------|
| PNSR↑ | Imuge+ | 24.11 | 23.02 | 23.88 | 23.80 | 22.46 | 23.84 | 23.49 | 23.53 | 20.25 |
| | W-RAE † | 26.63 | 25.77 | 18.44 | 10.81 | 10.16 | 26.36 | 22.41 | 25.16 | 24.19 |
| | ReImage | **31.91** | **30.21** | **29.69** | **28.94** | **27.97** | **31.74** | **28.68** | **29.09** | **27.87** |
| SSIM↑ | Imuge+ | 0.69 | 0.63 | 0.62 | 0.59 | 0.56 | 0.68 | 0.61 | 0.56 | 0.65 |
| | W-RAE† | 0.72 | 0.68 | 0.50 | 0.21 | 0.10 | 0.71 | 0.66 | 0.72 | 0.70 |
| | ReImage | **0.91** | **0.83** | **0.89** | **0.87** | **0.85** | **0.90** | **0.90** | **0.89** | **0.88** |
| LPIPS↓ | Imuge+ | 0.19 | 0.25 | 0.24 | 0.26 | 0.28 | 0.21 | 0.25 | 0.29 | 0.21 |
| | W-RAE† | 0.23 | 0.27 | 0.53 | 0.66 | 0.66 | 0.24 | 0.33 | 0.24 | 0.25 |
| | ReImage | **0.13** | **0.18** | **0.16** | **0.20** | **0.21** | **0.14** | **0.18** | **0.13** | **0.14** |

RAE (Cao et al., 2024b) does not use common degradations during training, we evaluate both the original version and a variant, retrained with common degradations.

In Table 2, our method consistently achieves the highest recovery performance across all tampering types. Compared to the best-performing baselines, it achieves an PSNR improvement of 29%, 29% and 23% in SD Inpaint, SDXL and Splicing, respectively. These results are further supported by the qualitative examples shown in Figure 4, where prior method often struggle to accurately localize tampered regions, leading to incomplete or low-fidelity recovery. Even when localization is successful, restoration quality remains limited. Additionally, in row 3, W-RAE fails to recover the original content when shuffled regions are also attacked, as the corresponding information has already been lost. In contrast, our method achieves significantly improved recovery results regardless of the attack. Due to the inherent trade-off between preserving the quality of the container image and maximizing the quality of the recovered image, our container images exhibit slightly lower SSIM and LPIPS values compared to W-RAE. However, this degradation is marginal considering the substantial improvement in recovery performance. Notably, our method also achieves the highest PSNR among all methods, including for the container image. Additional results with baseline models on diverse datasets and attacks are provided in the Appendix.

**Robustness against Common Degradation** To evaluate the robustness of ReImage, we conduct experiments under various common degradation settings, including Gaussian Noise (G.N.), JPEG Compression (JPEG), Gaussian Filter (G.F.), Median Filter (M.F.), Poisson Noise (P.N.), Hue Adjustment, Brightness Adjustment and Contrast Adjustment, as shown in Table 3. The Clean setting refers to the absence of any degradation. ReImage consistently shows significant improvements over the best-performing baselines, Imuge+ (Ying et al., 2023) and W-RAE (Cao et al., 2024b) across all degradation types, even when W-RAE is trained with common degradations. Compared to the Clean setting, performance shows a slight degradation across common distortions, with an average drop of 2.5 dB in PSNR, 0.035 in SSIM, and an increase of 0.04 in LPIPS. Nevertheless, our model demonstrates strong recovery capability, with the performance under Poisson Noise remaining nearly comparable to the Clean case. Details of common degradations and additional robustness results are provided in the Appendix.

## 5 CONCLUSION

In this paper, we presented ReImage, a neural watermarking-based image self-recovery framework that embeds a shuffled, high-frequency suppressed version of the original image into itself. By integrating four key components—Invertible Watermarking (IW) module, Watermark Generator (WG), Image Enhancement (IE) module, and Tamper Localization (TL) module—our method enables accurate and resilient recovery of tampered regions. Extensive experiments demonstrate that ReImage consistently outperforms existing approaches in terms of both the visual quality of the container and recovered images. This capability of ReImage is crucial for protecting users from manipulated content and enhancing the reliability of visual media.

## LLM USAGE

We used large language models (LLMs) solely as writing assistants to improve grammar, clarity, and readability. No part of the research ideation, experimental design, or substantive content relied on LLMs.

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

## A  ETHICS STATEMENT

Our work addresses ethical concerns around the authenticity of AI-generated and tampered media. ReImage is designed to restore original content rather than generate new content, thereby reducing risks of misuse common in generative models. Its primary social impact is in enhancing trustworthiness of visual media, mitigating misinformation, and supporting fair evaluation of digital evidence in domains such as journalism, law, and public communication.

## B  MORE DETAILS

### B.1  COMMON DEGRADATION

#### B.1.1  TRAINING

In this section, we detail the common degradations applied during training. Specifically, we implement four typical types of degradation: Gaussian Noise, JPEG Compression, Gaussian Filter, Median Filter. The details of each degradation are as follows:

- **Gaussian Noise** A Gaussian-distributed noise with a randomly selected standard deviation $\sigma \in [1, 16]$ is added to the container image.
- **JPEG Compression** A differentiable JPEG compression is applied to the container image, with the quality factor $Q \in [75, 95]$.
- **Gaussian Filter** A Gaussian smoothing filter with a randomly selected kernel size $k = 3$ and standard deviation $\sigma = 1.0$ is applied to the container image.
- **Median Filter** A median filter with a randomly selected kernel size $k = 3$ is applied to the container image to simulate nonlinear smoothing effects.

#### B.1.2  EVALUATION

In this section, we detail the common degradations applied during evaluation. Specifically, we implement eight typical types of degradation: Gaussian Noise, JPEG Compression, Poisson Noise, Gaussian Filter, Median Filter, Hue Adjustment, Brightness Adjustment and Contrast Adjustment. The details of each degradation are as follows:

- **Gaussian Noise** A Gaussian-distributed noise with a randomly selected standard deviation $\sigma \in [1, 9]$ is added to the container image.
- **JPEG Compression** A differentiable JPEG compression is applied to the container image, with the quality factor $Q = 90$.
- **Poisson Noise** A Poisson-distributed noise with an intensity parameter $\alpha = 4$ is added to the container image.
- **Gaussian Filter** A Gaussian smoothing filter with a randomly selected kernel size $k = 3$ and standard deviation $\sigma = 1.0$ is applied to the container image.
- **Median Filter** A median filter with a randomly selected kernel size $k = 3$ is applied to the container image to simulate nonlinear smoothing effects.
- **Hue Adjustment** The hue of the container image is adjusted by a random shift $\Delta h \in [-0.1, 0.1]$ in HSV color space to simulate color perturbations.
- **Brightness Adjustment** The brightness of the container image is adjusted by a random scaling factor $\beta \in [0.9, 1.1]$.
- **Contrast Adjustment** The contrast of the container image is modified by a random scaling factor $\gamma \in [0.7, 1.3]$.

### B.2  MASKING STRATEGY

As illustrated in Figure 5, we adopt two masking strategies from Sander et al. (2025) during training—Irregular Masking and Box-shaped Masking—to simulate tampering attacks. For each training

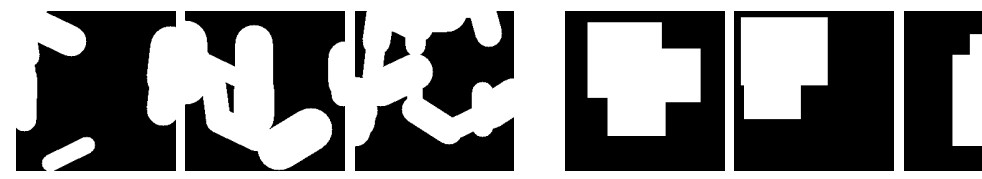

(a) Irregular Masking           (b) Box-shaped Masking

Figure 5: **Masking Strategy** (a) Irregular masking is implemented using random brush strokes and (b) box-shaped masking produces rectangular masked regions.

Table 4: **Computational Cost Comparison of Different Methods.** We compare the number of parameters (Param), floating-point operations (FLOPs), memory usage, and multiply–add operations (MAdd) across representative self-recovery models.

| Methods | Param | FLOPs | Memory | MAdd |
|---|---|---|---|---|
| W-RAE | 1.11M | 18.35G | 557MB | 18.42G |
| Imuge | 13.3M | 90.52G | 969MB | 90.37G |
| Imuge+ | 32.0M | 190.88G | 1580MB | 191.03G |
| ReImage | 9.94M | 64.76G | 4570MB | 36.44G |

sample, one of the two strategies is randomly selected with equal probability (*i.e.*, 50%), and a sampled region is replaced with a randomly chosen image. The details of each masking strategies are as follows:

- **Irregular Masking** uses random brush strokes, where parameters such as stroke angle (up to 4 degrees), length, and width (ranging from 20 to 50 pixels) are varied. Each image is overlaid with between 1 and 5 such strokes.

- **Box-shaped Masking** generates rectangular regions with a fixed 10-pixel margin. The dimensions of each box are randomly sampled between 50 and 150 pixels for both width and height, and 1 to 3 boxes are applied per image.

### B.3 ADDITIONAL IMPLEMENTATION DETAILS

We use an NVIDIA RTX 3090 Ti (24GB) for training and evaluation. The watermarking stage requires approximately 46 ms, and the self-recovery stage takes 166 ms. Thus, the total time for a full pipeline execution is approximately 212 ms.

We apply the discrete wavelet transform (DWT) and its inverse (IWT) with the Haar wavelet to the input and output of the Invertible Watermarking (IW) modules, respectively. DWT changes the image shape from $I \in \mathbb{R}^{H \times W \times 3}$ to $I \in \mathbb{R}^{\frac{H}{2} \times \frac{W}{2} \times 12}$ and this transformation is reversed by IWT. The transformer encoder within the Watermark Generator (WG) consists of multi-head self-attention layers with 6 heads and a token embedding dimension of 192. The Image Enhancement (IE) modules adopt the configuration from the original paper (Liu et al., 2025), with an upscale factor of 1, indicating that the resolution remains unchanged.

### B.4 COMPUTATIONAL COST

We compare the computational cost of ReImage and baseline models in terms of parameter size (Param), FLOPs, memory usage (Memory), and multiply–add operations (MAdds), all measured with an input image size of $256 \times 256$. As shown in Table 4, ReImage achieves lower FLOPs (64.76G) and MAdd (36.44G) compared to Imuge and Imuge+, while requiring fewer parameters overall. This indicates that ReImage is computationally more efficient in practice. Although the memory usage of ReImage (4570MB) is relatively higher than that of baseline methods, ReImage runs smoothly on widely accessible mid-range GPUs such as the RTX 3090 or even RTX 2080, demonstrating its practical deployability in real-world applications.

Table 5: **Comparison of Self-Recovery Methods under Various Levels of Noise.** We evaluate model robustness by measuring the quality of recovered images across varying levels of Gaussian Noise, JPEG Compression and Gaussian Filter. Note that † denotes models retrained with these degradations.

| Metrics | Methods | Gaussian Noise | | | JPEG Compression | | | | Gaussian Filter | |
|---|---|---|---|---|---|---|---|---|---|---|
| | | $\sigma = 1$ | $\sigma = 3$ | $\sigma = 5$ | QF = 60 | QF = 70 | QF = 80 | QF = 90 | $k = 3$ | $k = 5$ |
| PNSR↑ | Imuge+ | 24.02 | 23.42 | 22.85 | 22.87 | 23.19 | 23.53 | 23.88 | 23.80 | 23.53 |
| | W-RAE † | 26.60 | 26.30 | 25.76 | 16.10 | 17.13 | 17.95 | 18.44 | 10.81 | 11.63 |
| | ReImage | **31.85** | **31.29** | **30.13** | **27.10** | **27.81** | **28.61** | **29.07** | **28.94** | **27.36** |
| SSIM↑ | Imuge+ | 0.69 | 0.66 | 0.62 | 0.56 | 0.58 | 0.60 | 0.62 | 0.59 | 0.56 |
| | W-RAE † | 0.72 | 0.70 | 0.67 | 0.33 | 0.39 | 0.44 | 0.5 | 0.21 | 0.25 |
| | ReImage | **0.91** | **0.88** | **0.82** | **0.83** | **0.85** | **0.86** | **0.88** | **0.87** | **0.84** |
| LPIPS↓ | Imuge+ | 0.20 | 0.22 | 0.26 | 0.29 | 0.27 | 0.26 | 0.24 | 0.26 | 0.29 |
| | W-RAE † | 0.24 | 0.25 | 0.27 | 0.61 | 0.59 | 0.57 | 0.53 | 0.66 | 0.66 |
| | ReImage | **0.13** | **0.15** | **0.19** | **0.23** | **0.22** | **0.20** | **0.16** | **0.20** | **0.24** |

Table 6: **Comparison of Self-Recovery Methods under Common Geometric Distortions.** We evaluate the robustness of ReImage against three types of geometric distortions: Cropping, Rotation, and Resizing. Across all cases, ReImage consistently achieves higher PSNR performance compared to Imuge+.

| Methods | Crop | Rotate | Resize |
|---|---|---|---|
| Imuge+ | 22.58 | 20.12 | 24.81 |
| ReImage | 23.81 | 22.61 | 30.21 |

## C  ADDITIONAL EXPERIMENTAL RESULTS

### C.1  RECOVERY QUALITY UNDER VARYING LEVEL OF NOISE

To evaluate the robustness of ReImage, we additionally conduct experiments under varying levels of distortion. Specifically, we evaluated our method under JPEG compression (QF $\in 60, 70, 80, 90$), Gaussian noise ($\sigma \in 1, 3, 5$), and Gaussian filter ($k \in 3, 5$). As shown in Table 5, the performance of all models degrades as distortions become more severe. Nevertheless, ReImage consistently outperforms the baselines across the entire range of settings. Under Gaussian noise, the PSNR of all models decreases as the standard deviation increases. However, ReImage consistently outperforms both Imuge+ and W-RAE by 8dB and 5dB, respectively, across all noise levels. A similar trend is observed under Gaussian Filter, where the baselines show lower reconstruction quality at larger kernel sizes, while ReImage continues to achieve higher PSNR and SSIM. In the case of JPEG Compression, even at the lowest quality factor (QF = 60), ReImage attains superior PSNR, SSIM, and LPIPS compared to baselines, and its results at QF = 60 already surpass those of the baselines at QF = 90. These results indicate that although performance inevitably declines with stronger distortions, ReImage retains a consistent advantage over the baselines, achieving higher fidelity and perceptual quality across all tested settings.

### C.2  GEOMETRIC COMMON DEGRADATION

To further demonstrate the robustness of ReImage under diverse degradations, we conduct experiments on three geometric distortions: Cropping, Rotation and Resizing, evaluated using PSNR. For this setting, both ReImage and Imuge+ are additionally fine-tuned with geometric distortion types, where Imuge+ is chosen as the strongest-performing baseline. As shown in Table 6, ReImage consistently outperforms Imuge+ across all geometric distortions, with relative improvements of 5.4% for cropping, 12.4% for rotation, and 21.8% for resizing. Although the performance under geometric distortions is generally lower than that observed under valuemetric distortions such as noise or compression, ReImage still achieves clear gains over the baseline. Notably, under the Resizing, ReImage attains a performance level comparable to that achieved under valuemetric distortions.

Table 7: **Comparison of Self-Recovery Methods across Different Datasets.** We further evaluate the methods on additional datasets, including MS-COCO (Lin et al., 2014), CelebA-HQ (Zhu et al., 2022), and ILSVRC (Russakovsky et al., 2015). M-PSNR denotes the PSNR of the recovered image computed only over the tampered regions.

| Methods | MS-COCO | | CelebA-HQ | | ILSVRC | |
|---|---|---|---|---|---|---|
| | PSNR | M-PSNR | PSNR | M-PSNR | PSNR | M-PSNR |
| Imuge+ | 23.59 | 14.91 | 24.27 | 18.87 | 24.91 | 18.03 |
| W-RAE$^\dagger$ | 23.61 | 20.98 | 22.99 | 20.22 | 21.28 | 21.36 |
| ReImage | 30.57 | 23.93 | 29.33 | 24.08 | 29.93 | 23.50 |

Table 8: **Comparison of Tamper Localization Performance across Self-Recovery Methods** We simulate tampering using three methods: SD Inpaint (Rombach et al., 2022), SDXL (Podell et al., 2023), and splicing. Tamper localization performance is evaluated on the valAGE-Set using IoU, F1, and AUC metrics.

| Models | Tamper Localization | | | | | | | | |
|---|---|---|---|---|---|---|---|---|---|
| | SD Inpaint | | | SDXL | | | Splicing | | |
| | IoU↑ | F1↑ | AUC↑ | IoU↑ | F1↑ | AUC↑ | IoU↑ | F1↑ | AUC↑ |
| Imuge | 0.61 | 0.69 | 0.90 | 0.60 | 0.68 | 0.90 | 0.61 | 0.69 | 0.90 |
| Imuge+ | 0.20 | 0.25 | 0.89 | 0.24 | 0.25 | 0.89 | 0.62 | 0.69 | 0.97 |
| W-RAE$^\dagger$ | 0.49 | 0.61 | 0.71 | 0.49 | 0.61 | 0.72 | 0.50 | 0.61 | 0.73 |
| ReImage | **0.82** | **0.90** | **0.99** | **0.82** | **0.91** | **0.99** | **0.84** | **0.93** | **0.99** |

## C.3 Ablation Study on Various Dataset

We conduct additional experiments on diverse datasets to assess the generalization ability of ReImage. Specifically, we use CelebA-HQ (Zhu et al., 2022) and ILSVRC (Russakovsky et al., 2015). Tampering is simulated with Stable Diffusion Inpaint (SD Inpaint). We generate masks to explicitly define the tampered regions. For CelebA-HQ, we create masks based on facial attributes, and for ILSVRC, we employed SAM (Kirillov et al., 2023) to obtain object-level masks. For evaluation, we randomly sample the same number of images as reported in Imuge+ (520 for CelebA and 1,047 for ILSVRC).

As shown in Table 7, ReImage achieves the highest performance across all datasets, consistently outperforming Imuge+ and W-RAE. On CelebA-HQ, ILSVRC, and MS-COCO, our model yields on average over 20% improvement in PSNR and 15% in M-PSNR compared to the baselines. Notably, the performance is consistent across datasets, indicating no significant gap between them. These results demonstrate that our model generalizes robustly and effectively to diverse datasets used in prior studies.

## C.4 Tamper Localization

To evaluate the effectiveness of the Tamper Localization (TL) module in ReImage, we compare it against recent self-recovery methods, including Imuge (Ying et al., 2021), Imuge+ (Ying et al., 2023), and W-RAE (Cao et al., 2024b). We simulate tampering using Stable Diffusion Inpainting (Rombach et al., 2022), Stable Diffusion XL (Podell et al., 2023), and splicing attacks, guided by mask annotations from the valAGE-Set. Localization accuracy is measured using Intersection over Union (IoU), F1 score, and Area Under the ROC Curve (AUC).

As shown in Table 8, ReImage consistently achieves the highest tamper localization performance across various tampering types. Compared to Imuge (Ying et al., 2021), the strongest baseline among existing methods, ReImage achieves relative improvements of 34%, 30%, and 10% in IoU, F1, and AUC, respectively. Furthermore, while Imuge+ (Ying et al., 2023) performs poorly on unseen attacks such as SD Inpaint and SDXL, ReImage remains effective. This demonstrates that ReImage can accurately localize tampered regions even under unseen or diverse attack types.

Table 9: **Additional Comparison of Self-Recovery Methods on valAGE-Set** We evaluate all baselines using M-PSNR to assess recovery quality specifically within tampered regions, reporting results under both predicted and GT mask settings. "M-PSNR" denotes the PSNR of the recovered image, computed only over the tampered regions. The "GT mask" setting uses ground-truth tampering masks instead of predicted ones.

| Models | Recovered Image $\hat{I}_{\text{rec}}$ | | | |
| | SD Inpaint | SDXL | Splicing | GT mask |
| | M-PSNR↑ | M-PSNR↑ | M-PSNR↑ | M-PSNR↑ |
|---|---|---|---|---|
| Imuge | 10.66 | 10.59 | 10.61 | 11.23 |
| Imuge+ | 14.91 | 14.86 | 16.54 | 22.37 |
| W-RAE$^{\dagger}$ | 20.98 | 21.00 | 19.86 | 21.64 |
| ReImage | **23.93** | **23.96** | **22.59** | **25.95** |

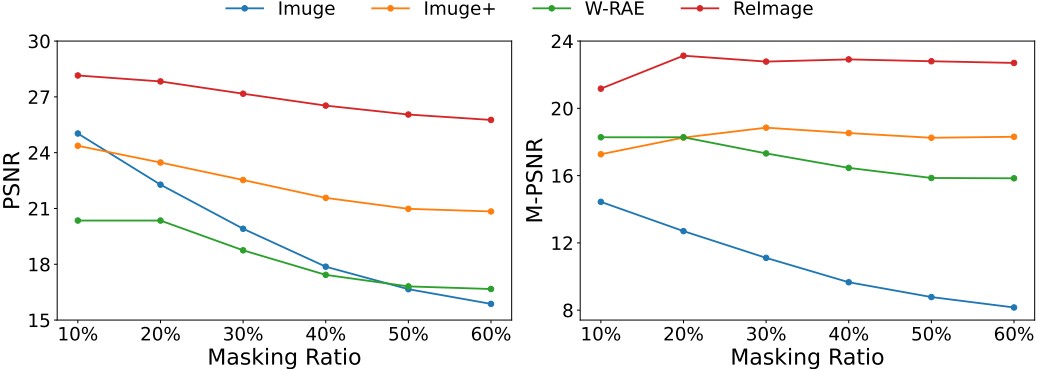

Figure 6: **Comparison of Self-Recovery Methods Across Various Masking Ratios.** We evaluate the recovery performance of all baseline methods under different masking ratios, ranging from 10% to 60%. Tampering is simulated via splicing attacks, where tampered regions are randomly generated using a combination of circles and rectangles.

## C.5 RECOVERY QUALITY IN TAMPERED REGIONS

We conduct additional experiments to evaluate image self-recovery performance on the valAGE-Set, comparing ReImage with recent state-of-the-art methods, including Imuge (Ying et al., 2021), Imuge+ (Ying et al., 2023), and W-RAE (Cao et al., 2024b). Specifically, we evaluate the M-PSNR across the same tampering types as in Table 8. As shown in Table 9, ReImage consistently achieves the highest performance, with relative improvements of 14%, 14%, and 14% over W-RAE. These results demonstrate the effectiveness of our approach in recovering tampered regions across diverse types of attacks.

We further evaluate all methods under the ground-truth (GT) mask setting, where ground-truth tampering masks are used in place of predicted ones. This setting isolates the effect of recovery performance from localization accuracy. As shown in Table 9, ReImage achieves the highest performance, outperforming Imuge+ by a relative margin of 16% in M-PSNR. Notably, ReImage outperforms both Imuge+ and W-RAE, even though they are provided with GT masks, whereas ReImage operates without them. These results highlight the effectiveness of our pipeline not only in self-recovery but also in tamper localization. Moreover, these results suggest that ReImage could achieve even greater performance with improved tamper localization accuracy.

## C.6 RECOVERY QUALITY UNDER VARIOUS MASKING RATIOS

To evaluate the robustness of our method under varying tampering intensities, we conduct experiments across different masking ratios. Specifically, we compare our approach with recent state-of-the-art self-recovery methods. Tampering is simulated via splicing attacks, where each mask contains 1 to 3 randomly placed geometric shapes (rectangles or circles). The masking ratio is var-

Table 10: **Additional Comparison of Self-Recovery Methods under Photoshop Editing.** We evaluate recovery quality of all baselines under Photoshop editing operations to assess robustness against realistic manipulations. "M-PSNR" denotes the PSNR of the recovered image, computed only over the tampered regions.

| Models | PSNR | SSIM | LPIPS | M-PSNR |
|---|---|---|---|---|
| Imuge+ | 21.13 | 0.62 | 0.24 | 13.50 |
| W-RAE$^\dagger$ | 24.94 | 0.68 | 0.28 | 20.70 |
| ReImage | **28.85** | **0.88** | **0.15** | **22.75** |

Table 11: **Additional Study on the Impact of Core Components in ReImage** We examine the contribution of the core components-Pixel Shuffling (PS), Image Enhancement (IE), and the Watermark Generator (WG)-by evaluating all possible combinations. Using the full set of components consistently yields the best performance, both in terms of container and recovered image quality. M-PSNR denotes the PSNR of the recovered image computed only over the tampered regions.

| PS | IE | WG | Container Image PSNR↑ | Recovered Image PSNR↑ | M-PSNR↑ |
|---|---|---|---|---|---|
| ✗ | ✗ | ✗ | 35.93 | 25.82 | 16.44 |
| ✓ | ✗ | ✗ | 25.78 | 22.95 | 18.32 |
| ✗ | ✓ | ✗ | 41.28 | 24.21 | 13.80 |
| ✗ | ✗ | ✓ | 42.24 | 23.53 | 13.61 |
| ✓ | ✓ | ✗ | 26.19 | 24.25 | 18.91 |
| ✗ | ✓ | ✓ | 39.40 | 26.47 | 16.96 |
| ✓ | ✗ | ✓ | 33.17 | 29.27 | 23.17 |
| ✓ | ✓ | ✓ | 36.10 | 30.57 | 23.93 |

ied across 10%, 20%, 30%, 40%, 50%, and 60% of the total image area to reflect different levels of tampering. We report PSNR and Mask PSNR (M-PSNR) as evaluation metrics.

As shown in Figure 6, our method consistently achieves strong performance across all evaluation metrics and masking ratios. In particular, ReImage outperforms the best-performing baselines by more than 5dB in both PSNR and M-PSNR. Even at a masking ratio of 60%, ReImage achieves a higher PSNR than Imuge at 10%. Furthermore, while Imuge and W-RAE degrade significantly as the masking ratios increases, ReImage shows only a slight decrease of approximately 4dB and 0.39dB in PSNR and M-PSNR, respectively, between the 10% and 60% masking ratios. demonstrating robustness to tampering intensity.

### C.7 ABLATION STUDY ON PHOTOSHOP EDITING

We evaluate ReImage under realistic tampering scenarios. To simulate such manipulations, we employ Adobe Photoshop with the MS-COCO (Lin et al., 2014) dataset. Specifically, we select objects within images and apply removal or inpainting operations using Photoshop's integrated generative model. A total of 100 attacked images are collected for evaluation. As shown in Table 10, ReImage achieves the highest recovery performance compared to the baseline models Imuge+ and W-RAE. Compared with the best-performing baseline, ReImage improves PSNR and M-PSNR by 3.91 dB and 2.05 dB, respectively. While the results under Photoshop editing are slightly lower than those in Table 2, our method still demonstrates strong recovery capability. Furthermore, as shown in Figure 8 qualitative results in clearly show that ReImage successfully restores tampered regions with high visual quality.

### D DISCUSSION

#### D.1 EFFECT OF EACH COMPONENT

Table 11 includes settings that are not presented in the main paper, allowing us to compare all combinations of PS, WG, and IE components when combined with the results in Table 1. This

Table 12: **Impact of WG Loss in ReImage.** We evaluate the contribution of WG Loss ($\mathcal{L}_{\mathrm{WG}}$). While using $\mathcal{L}_{\mathrm{IE}}$ alone provides lower recovery quality, combining it with $\mathcal{L}_{\mathrm{WG}}$ significantly enhances performance, demonstrating the importance of WG Loss in guiding more accurate image reconstruction.

| $\mathcal{L}_{\mathrm{WG}}$ | $\mathcal{L}_{\mathrm{IE}}$ | Container Image $I_{\mathrm{con}}$ | | | Recovered Image $\hat{I}_{\mathrm{rec}}$ | | | |
|:---:|:---:|:---:|:---:|:---:|:---:|:---:|:---:|:---:|
| | | PSNR↑ | SSIM↑ | LPIPS↓ | PSNR↑ | SSIM↑ | LPIPS↓ | M-PSNR↑ |
| ✗ | ✓ | 36.56 | 0.94 | 0.11 | 25.00 | 0.85 | 0.20 | 15.78 |
| ✓ | ✓ | 36.10 | 0.93 | 0.10 | 30.57 | 0.87 | 0.17 | 23.93 |

enables a more comprehensive analysis of each component's individual contribution. As shown in the Table 11, each component of our model contributes to improved image self-recovery performance. We observe consistent performance gains as more components are combined, with the best results achieved when all three (PS, WG, and IE) are used together. This indicates that each module contributes complementary benefits, rather than interfering with one another. Additionally, the relatively high quality of the container image without PS can be attributed to the model embedding less information, which in turn makes recovery more difficult.

## D.2    EFFECT OF WG LOSS

The loss in $\mathcal{L}_{\mathrm{WG}}$ is additionally introduced to encourage the Watermark Generator (WG) to retain some capability to recover the original image. To assess its necessity, we compare the performance of using only $\mathcal{L}_{\mathrm{IE}}$ versus using both $\mathcal{L}_{\mathrm{WG}}$ and $\mathcal{L}_{\mathrm{IE}}$. As shown in Table 12, we observe a clear improvement in the quality of the recovered image when both $\mathcal{L}_{\mathrm{WG}}$ and $\mathcal{L}_{\mathrm{IE}}$ are used. Specifically, $\mathcal{L}_{\mathrm{WG}}$ encourages the inverse process of WG to retain a coarsely recovered version of the original image, allowing the Image Enhancement (IE) module to focus only on restoring the remaining missing details. Without $\mathcal{L}_{\mathrm{WG}}$, however, the combined WG and IE behave more like a single large watermark decoder, lacking the structural separation between coarse recovery and refinement. This may prevent the model from leveraging the inductive bias of performing a rough watermark recovery before enhancement, leading to suboptimal performance.

## E    LIMITATIONS

ReImage consistently demonstrates high fidelity in both container and recovered images across various tampering methods, as verified through both qualitative and quantitative evaluations. Nonetheless, it exhibits limitations in certain scenarios. In particular, ReImage often fails to reconstruct tampered regions when geometric transformations such as rotation and cropping are applied, as these distort the embedded secret image and degrade the information necessary for accurate recovery.

In addition, tamper localization can impose constraints on performance. When the predicted region is smaller than the actual tampered area, parts of the attacked content may remain unrecovered, since the corresponding region from the container image—including tampered parts—is used to overwrite the recovered image. This may lead to an underestimation of ReImage's true capability.

## F    ADDITIONAL VISUAL RESULTS

In this section, we extend our evaluation by visualizing the robustness of ReImage against various common degradations (*i.e.*, Gaussian Noise (G.N.), JPEG Compression (JPEG), Gaussian Filter (G.F.), Median Filter (M.F.), Poisson Noise (P.N.), Hue Adjustment, Brightness Adjustment and Contrast Adjustment), as shown in Figure 7. These experiments show that ReImage remains effective under various content-preserving modifications. Additionally, we present further qualitative comparisons on the valAGE-Set between ReImage and other baseline methods. As shown in Figure 9 and Figure 10, ReImage consistently generates high-fidelity container and recovered images, even under tampering attacks and degradations. In contrast, other methods either fail in both aspects or struggle to balance imperceptibility and accurate recovery.

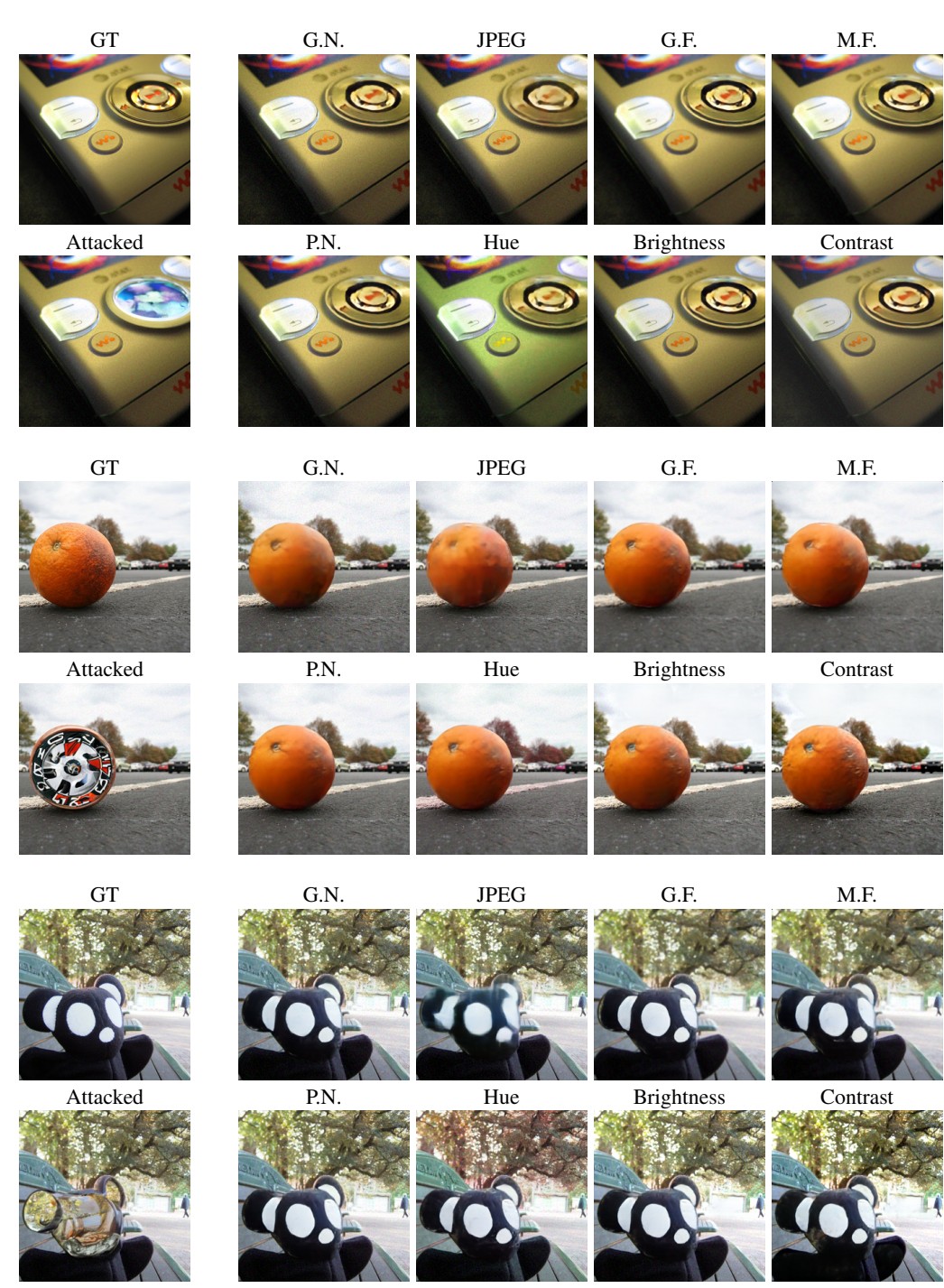

Figure 7: **Qualitative Results under Various Common Degradation Types** We qualitatively evaluate the robustness of ReImage under eight common degradation types—Gaussian Noise (G.N.), JPEG Compression (JPEG), Gaussian Filter (G.F.), Median Filter (M.F.), Poisson Noise (P.N.), Hue Adjustment, Brightness Adjustment and Contrast Adjustment.

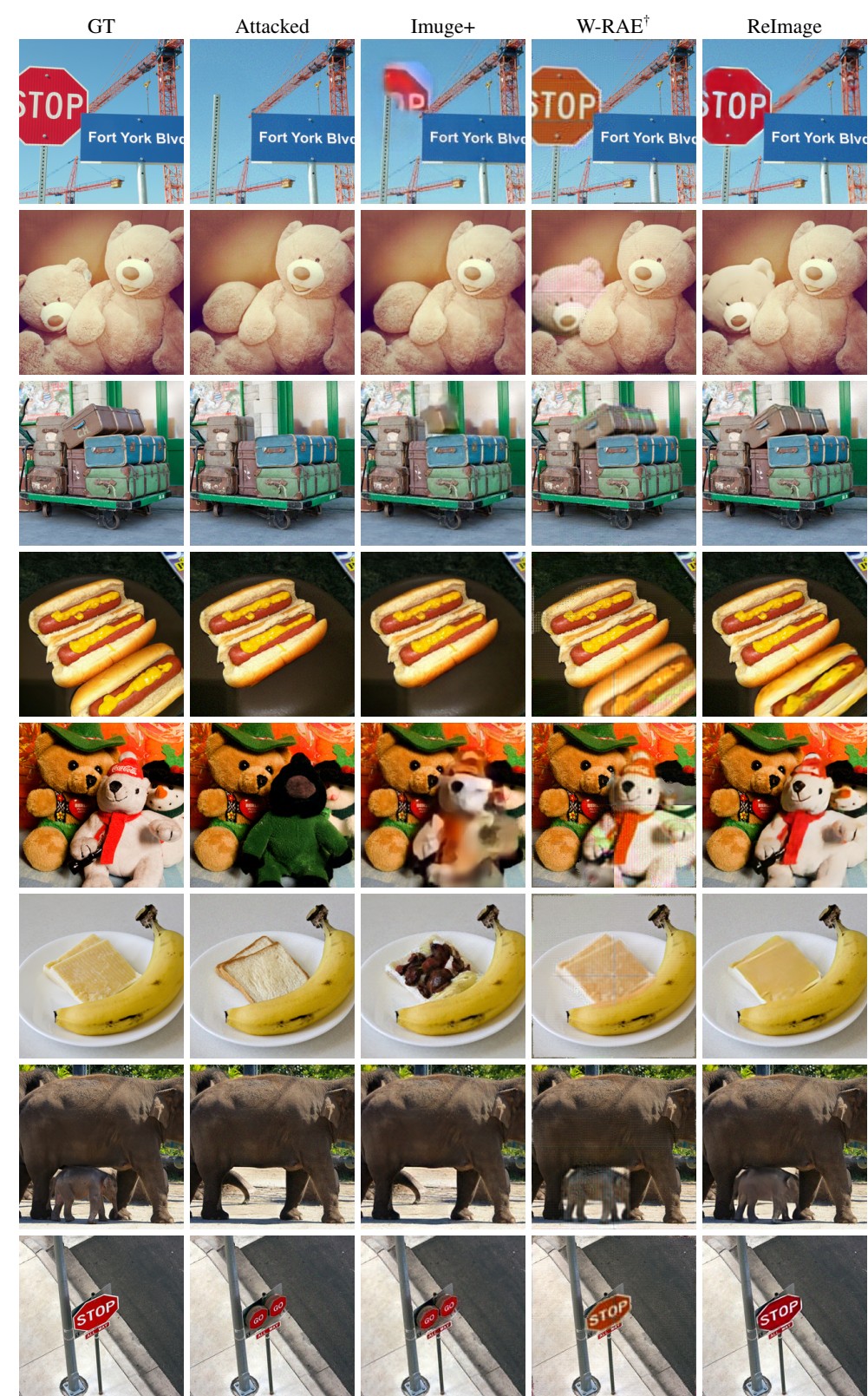

Figure 8: **Visual Comparison of Recovered Images from Different Self-Recovery Methods under Photoshop Editing** Tampering is simulated using Adobe Photoshop. ReImage demonstrates high-fidelity reconstruction of the original image under realistic manipulation

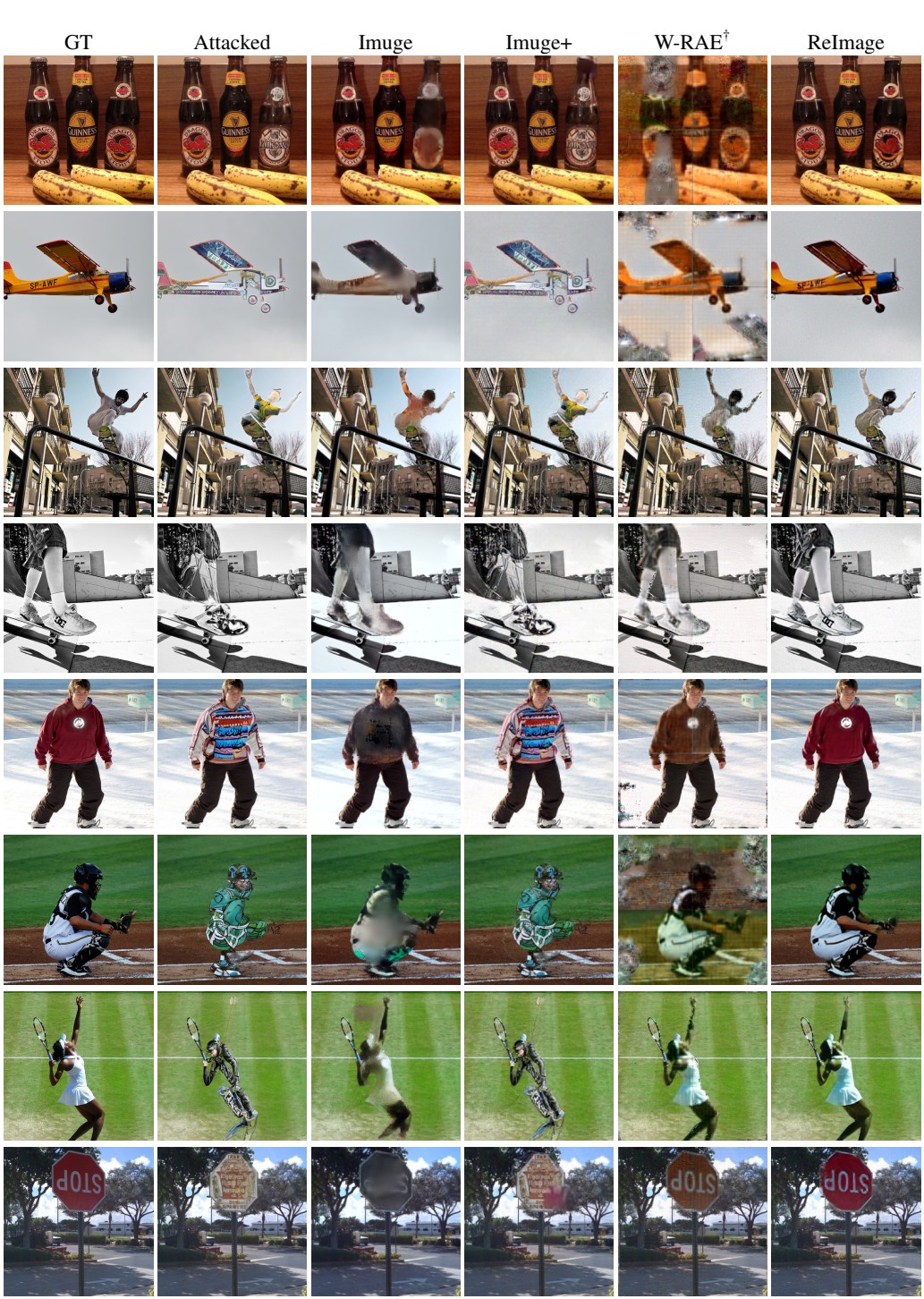

Figure 9: **Visual Comparison of Recovered Images from Different Self-Recovery Methods on valAGE-Set** Tampering is simulated using SD Inpaint (Rombach et al., 2022). ReImage demonstrates high-fidelity reconstruction of the original image.

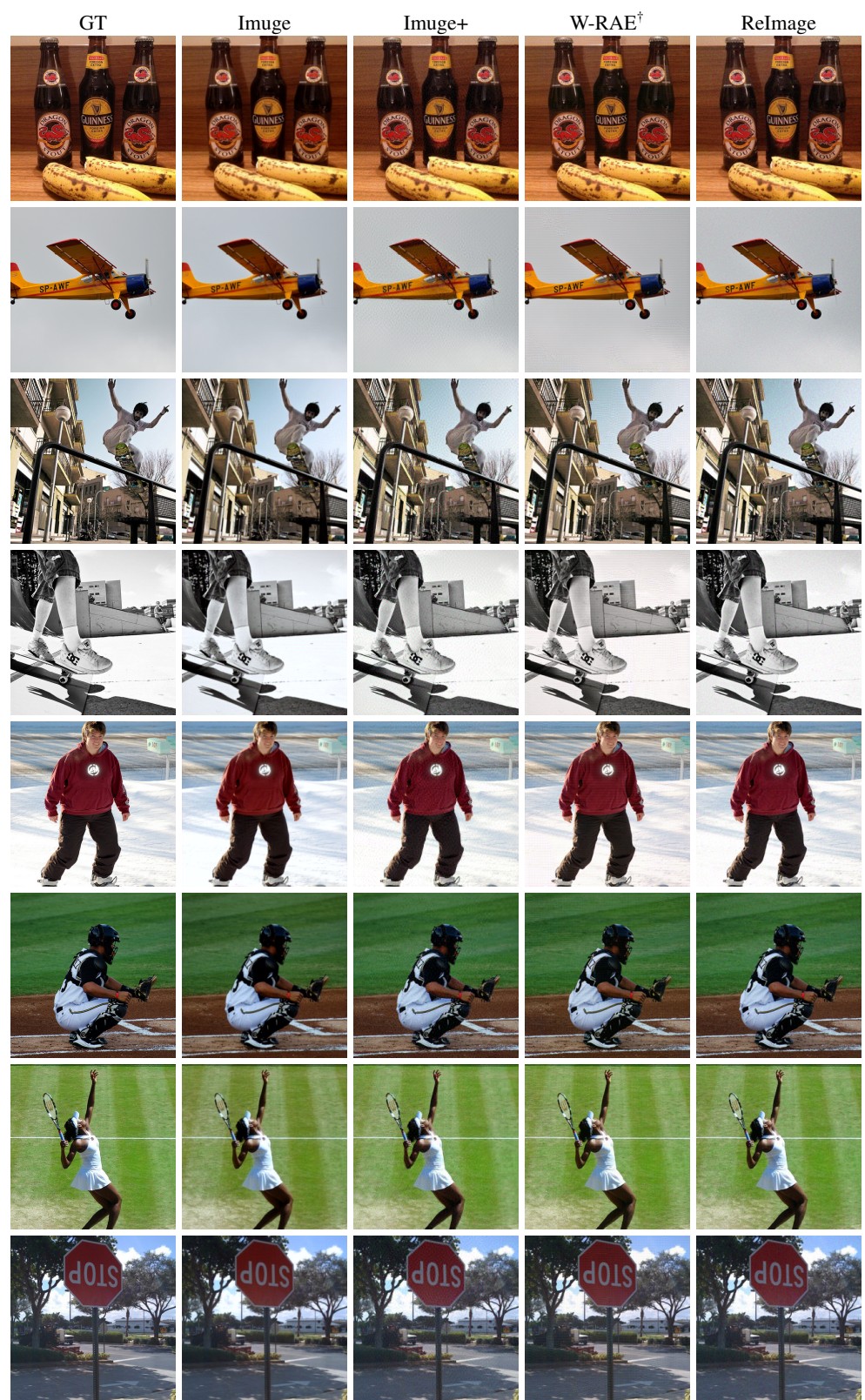

Figure 10: **Visual Comparison of Container Images from Different Self-Recovery Methods on valAGE-Set** Tampering is simulated using SD Inpaint (Rombach et al., 2022). ReImage demonstrates the ability to embed watermarks into the target image with minimal visual distortion.

