# OpenReview forum: "Robust Image Self-Recovery against Tampering using Watermark Generation with Pixel Shuffling"
_ICLR.cc/2026/Conference — ICLR 2026 Conference Withdrawn Submission_

### Official Review · Reviewer_w6yU · 2025-10-17

**Soundness:** 2
**Presentation:** 2
**Contribution:** 2
**Rating:** 2
**Confidence:** 4

**Summary:**

This submission aim to embed self-recovery watermarks into images to enable tamper localization and original content reconstruction. This work advances prior work by addressing spatial alignment fragility via fine-grained pixel shuffling, introducing a learned Watermark Generator (WG) to suppress high-frequency artifacts, and adding an Image Enhancement (IE) module.

**Strengths:**

Results show that the recovery quality of the method is higher. In many tests, the results are much better than the existing methods.

**Weaknesses:**

1. It seems the applied validation dataset is different from the applied baselines. It seems how these baseline models are prepared remain unclear.

2. It seems the applied methodology lacks novelty. Like Invertible networks and mask free generation are also applied in Imuge.

3. It seems the reviewer cannot easily benchmark the advantage of this method, either via theoretical or empirical analysis, or via source code (no code or API provided. though it is completely optional, the reviewer cannot play with the model to address some concerns).

4. It seems the role of accurate attack localization is underestimated (why does the propose mechanisms also boost localization? And why the compared methods fail in the detection stage in many provided samples?) The review can only see the lower results of the compared methods via figures and tables, but barely see the performance of attack localization (except table 8, deeply hidden in the supplement). The reviewer suspects that the focus on localization training of the proposed method more contributes to the final result (especialy considering the reported large performance gap). The reviewer think in this task, authors need to report fair quantitative comparison of recovery using same localization result (either being it a good one or a poor one), in order to benchmark the contribution of modules like WG, IE, etc. In the current version, the two factors are simply interwined.

5. The reviewer thinks the visualization of the proposed method can be rather biased. The authors do show the improvement in the end-to-end result (better recovered images). However, the reviewer barely see any visualization of intermediate results, i.e., prediction mask, or any other signals that could somehow interpret the successes. The authors claim that the compared baseline methods can be less robust or show less generalizability. However, in many shown cases, the baselines simply cannot locate the attack, which can significantly lower their overall performance. So the reviewer personally is in favor of a more well-rounded result report than a biased one.

6. Figure 6 reports results under simulated attacks, which can be less convincing, since attacking images using circles and rectangles can easily leave traces for localization. Also, the. difference between table 7 and table 9 is worth inquiring: does table 7 report in-domain (train and test from a same dataset) performance and table 9 report cross-domain? If not, the reviewer cannot understand why there is a large gap of the baselne performances between the two tables. Also, the figure 10, the watermarks can be barely seen for all methods.

7. Finally, the baselines can be further improved. From the results, we see that the baselines have a gap with the proposed method. Meanwhile, imuge and imuge+ are proposed by a same team, and W-RAE is also using INN for this task. Thus, the reviewer think we need to add more stronger baselines. Remind: it seems this task can be easily achieved by many other image translation networks, such as transformers, diffusions, mambas, etc.

**Questions:**

Please refer to the previous section.

**Details Of Ethics Concerns:**

n.a.

---

### Official Review · Reviewer_ZhA1 · 2025-10-17

**Soundness:** 2
**Presentation:** 3
**Contribution:** 2
**Rating:** 2
**Confidence:** 4

**Summary:**

This paper focuses on image self recovery, which in short is to locate the tampered areas within an image and try to recover the original area. The proposed method ReImage proposes embedding a shuffled version of the target image as a watermark via a novel framework to misalign corrupted regions and their watermarks.

**Strengths:**

- Authors design several modules to improve the self-recovery, and the recovered images are better.
- Authors provides many experimental results to show the performance in different aspects.

**Weaknesses:**

- This paper to me is more like an engineering paper. Many parts (from the pipeline: two-staged, similar to the compared imuge, to the model: inn, similar to W-RAE, again to the training details: mixed and fixed jpeg, filtering, etc.) show minor academic innovation. The proposed pixel shuffling for watermark and the two modules (IE and WG) are also trivial designs.
- Pixel shuffling improved image recovery, but also increase the entropy of the information to be hidden. After reading Section 3.3.2, I still don't understand why in the proposed method, both imperceptibility and recovery are both improved (simply via using transformer blocks in INN?)
- Experimental details seems lacking, and the compared results deviate from the ones reported in the paper.

**Questions:**

Major:
- How are the methods compared? I see a significant performance gap between the original reported ones and those in here. E.g., In W-RAE, the averaged reported PSNR was 32.42 (db), while in this paper mostly lower than 25db.
- Why do the authors use a novel dataset setting for comparison? Rather than accepting an existing experiment setting? Why a third party paper Editguard's criterion is applied here?
- In the tables and figures i don't see the mask result. Also in table 9 the psnr of two compared method are as low as 15- db. Authors should give mandatory test details to indicate that this was not a mistake.
- Ensuring weak data hiding and good performance is a trade-off. Just like what is mentioned in "weaknesses", still do not understand how is the method so efficient in the watermarking? Also, what block size do you use in the experiments? If i understand it corrent, the P=4 refers to the num of block in transformer? Besides, why the curve of recovered image also go down with the increase of num of blocks? WIth hidden image (ground truth) shuffled further, we should expect that the recovery capacity should increase, despite the burden on the embedding side.
- Unclear motivation: "Since we apply the shuffling algorithm, the recovered image ˆIorg exhibits globally distributed degradation" i

Minor:
- Real-scene attacks can be more likely compound attacks than rare attacks like hue adjust and contrast change. E.g., dual compression, or rescale than JPEG.
- It is not easy for me to distinguish which table and figures are reporting results on real-scene attacks, which are on simulated conditions.
- Writing issue - you should refrain from always using the right (closing) quotation mark in the paper.

---

### Official Review · Reviewer_qAz2 · 2025-10-29

[review text omitted: it was posted to a different submission]

---

> ### Author Response · Authors · 2025-11-12
>
> I sincerely appreciate your comment.
>
> However, it appears that a review for a different paper was mistakenly posted under our submission.
>
> We would greatly appreciate it if you could kindly check once more to ensure it was posted correctly.

---

> > ### Comment · Reviewer_qAz2 · 2025-11-12
> > **Correction for the review comment**
> >
> > So sorry for the wrongly posted reviews, and I guess this one is correct now. I apologize for any inconvenience caused.

---

> > > ### Author Response · Authors · 2025-11-12
> > >
> > > We appreciate your quick correction.
> > >
> > > However, it seems that the Strength section is still for a different paper.
> > >
> > > We would sincerely appreciate it if you could kindly verify this once again.

---

> > > > ### Comment · Reviewer_qAz2 · 2025-11-13
> > > >
> > > > Sorry for the delayed correction. I have updated the Strength part.

---

### Official Review · Reviewer_ArYg · 2025-11-01

**Soundness:** 3
**Presentation:** 3
**Contribution:** 3
**Rating:** 8
**Confidence:** 4

**Summary:**

The paper introduces ReImage, a self-recovery framework for images based on neural watermarking. The approach embeds a shuffled, high-frequency-suppressed version of the target image into itself as a watermark using an invertible neural network (INN)-based system. The key contributions include a specialized watermark generator, pixel shuffling to disrupt spatial correlation for better tamper recovery, and an image enhancement module to further improve output fidelity. Extensive experiments on the MS-COCO2017 dataset and various tampering scenarios show state-of-the-art performance, both in visual quality and robustness to common degradations.

**Strengths:**

1. Novel and Effective Framework: The paper proposes ReImage, a well-designed neural watermarking-based self-recovery method that leverages pixel shuffling to spatially misalign watermark content with the image. This innovation addresses a known issue of recovery failure due to alignment between tampered and watermarked regions along with clustered tampered regions in secret image.

2. Thorough Design and Ablation Study: The architecture is modular and interpretable, consisting of components like the invertible watermarking network, a learned watermark generator, image enhancement, and tamper localization. Ablation experiments clearly demonstrate the impact of each component on recovery performance.

3. Superior Performance on Tampering Tasks: On various tampering types (e.g., SD-Inpaint, SDXL, and splicing), ReImage achieves state-of-the-art results, showing improvements over prior work like W-RAE and Imuge+.

**Weaknesses:**

1. Insufficient Evaluation under diverse degradations: While robustness to three types of degradations is briefly evaluated (Gaussian noise, JPEG compression, and Poisson noise), the degradation types are limited, and there is no geometric degradation included (Imuge has included cropping in its experimental evaluation). It is well known that geometric degradations, such as cropping, pose significant challenges for watermarking models, suggesting a potential trade-off between robustness and image quality.

2. Limited Real-World Validation: The limited range of tested degradations raises concerns about the method’s practicality, as real-world scenarios often involve more complex and compound distortions—such as slight cropping, minor rotations, or cases where a region of an image is cropped and spliced onto another image. These types of manipulations were not sufficiently explored, making it unclear how well the method generalizes beyond controlled experimental settings.

**Questions:**

Please refer to the weakness.

---

### Comment · Area_Chair_RQDf · 2025-11-13

Dear Authors,

Received your message and noted your concern. I think that reviewer qAz2 has corrected it. If you have any problem, please message me.

Regards,
AC

---

### Note · Authors · 2025-11-13

I have read and agree with the venue's withdrawal policy on behalf of myself and my co-authors.